# Comparison of patient-reported outcomes at one year after injury between limb salvage and amputation: A prospective cohort study

**Taketo Kurozumi****\*, Takahiro Inui, Yuhei Nakayama, Akifumi Honda, Kentaro Matsui, Keisuke Ishii, Takashi Suzuki, Yoshinobu Watanabe**

Trauma and Reconstruction Center, Teikyo University Hospital, Tokyo, Japan

\* taketo_kurozumi@m6.dion.ne.jp

## Abstract

### Purpose

This single-center, prospective cohort study aimed to compare the patient-reported outcomes one year after injury between limb salvage and amputation and to elucidate whether amputation contributes to early recovery of functionality and quality of life.

### Methods

We included 47 limbs of 45 patients with severe open fractures of the lower limb and categorized them into limb salvage and amputation groups. Data on patient-reported outcomes one year after injury were obtained from the Database of Orthopaedic Trauma by the Japanese Society for Fracture Repair at our center. Patients' limbs were evaluated using the lower extremity functional scale and Short-Form 8. Early recovery was assessed using functionality and quality-of-life questionnaires.

### Results

Of the 47 limbs, 34 limbs of 34 patients were salvaged, and 13 limbs of 11 patients were amputated. Significant differences were noted between the limb salvage and amputation groups in terms of the lower extremity functional scale scores (mean: 49.5 vs. 33.1, P = 0.025) and scores for the mental health component (mean: 48.7 vs. 38.7, P = 0.003), role–physical component (mean: 42.2 vs. 33.3, P = 0.026), and mental component summary (mean: 48.2 vs. 41.3, P = 0.042) of the Short-Form 8. The limb salvage group had better scores than the amputation group.

### Conclusions

As reconstruction technology has advanced and limb salvaging has become possible, the focus of studies should now be based on the perspective of "how the patient feels;" hence, we believe that the results of this study, which is based on patient-reported outcomes, are meaningful.

**Data Availability Statement:** Request can be made to the Japanese Society for Fracture Repair secretariat (1-8-6, Shintomi Chuoku Tokyo 104-

0041 Japan) or to jsfr@jsfr.jp and those who meet the criteria for access to anonymized patient level data will be granted data access.

**Funding:** The authors received no specific funding for this work.

**Competing interests:** The authors have declared that no competing interests exist.

# Introduction

## Background

Advances in knowledge and technology have made salvaging the limbs of patients with severe trauma and injuries possible. Some reports suggest that amputation and limb salvage results for these injuries are similar [1–3]. However, even in recent years, amputation is occasionally the chosen course of action, rather than limb salvage, in severe limb injuries because it allows patients to regain their social lives sooner [4–6]. Moreover, many previous investigations on this topic are retrospective cohort studies or meta-analyses on retrospective studies, whereas prospective cohort studies based on patient-reported outcomes, such as the LEAP study, are extremely rare [1–3].

## Objective

The present study aimed to compare the patient-reported outcomes, such as limb function and quality of life, at one year after injury between limb salvage and amputation in patients with Gustilo classification IIIb and IIIc fractures to determine whether amputation leads to early recovery and regaining of social lives.

# Materials and methods

## Study design

A single-center, prospective cohort study.

## Setting

The Trauma and Reconstruction Center, Teikyo University Hospital, where patients with orthopedic injuries and trauma are managed.

## Data source

The Database of Orthopaedic Trauma managed by the Japanese Society for Fracture Repair.

## Variable

Patient age, sex, Gustilo classification, AO Foundation/Orthopedic Trauma Association (AO/OTA) classification, Orthopedic Trauma Association Open Fracture Classification (OTA-OFC), OTA-OFC summative score [7], complications, pre-operative and post-operative lower extremity functional scale (LEFS) score [8], post-operative Short-Form 8 (SF-8) score [6], number of operations, surgical site infection rate, and patient employment status.

The study conformed to the principles of the Declaration of Helsinki and its amendments. Approval was granted by the Institutional Review Board of Teikyo University Ethical Review Board for Medical and Health Research Involving Human Subjects (approval number: 14-167-3). Written and verbal informed consent was obtained from all patients prior to treatment and participation.

## Participants

Of the 439 limbs of 414 registered patients with open long-bone fractures at our center between February 2015 and December 2019, 69 limbs of 65 patients were diagnosed with Gustilo classification IIIb and IIIc fractures, and 53 limbs of 51 patients were diagnosed with lower limb fractures, according to the AO/OTA classification 41–44. This study included 47 limbs of 45 patients diagnosed as Gustilo classification IIIb and IIIc as well as lower limb fractures

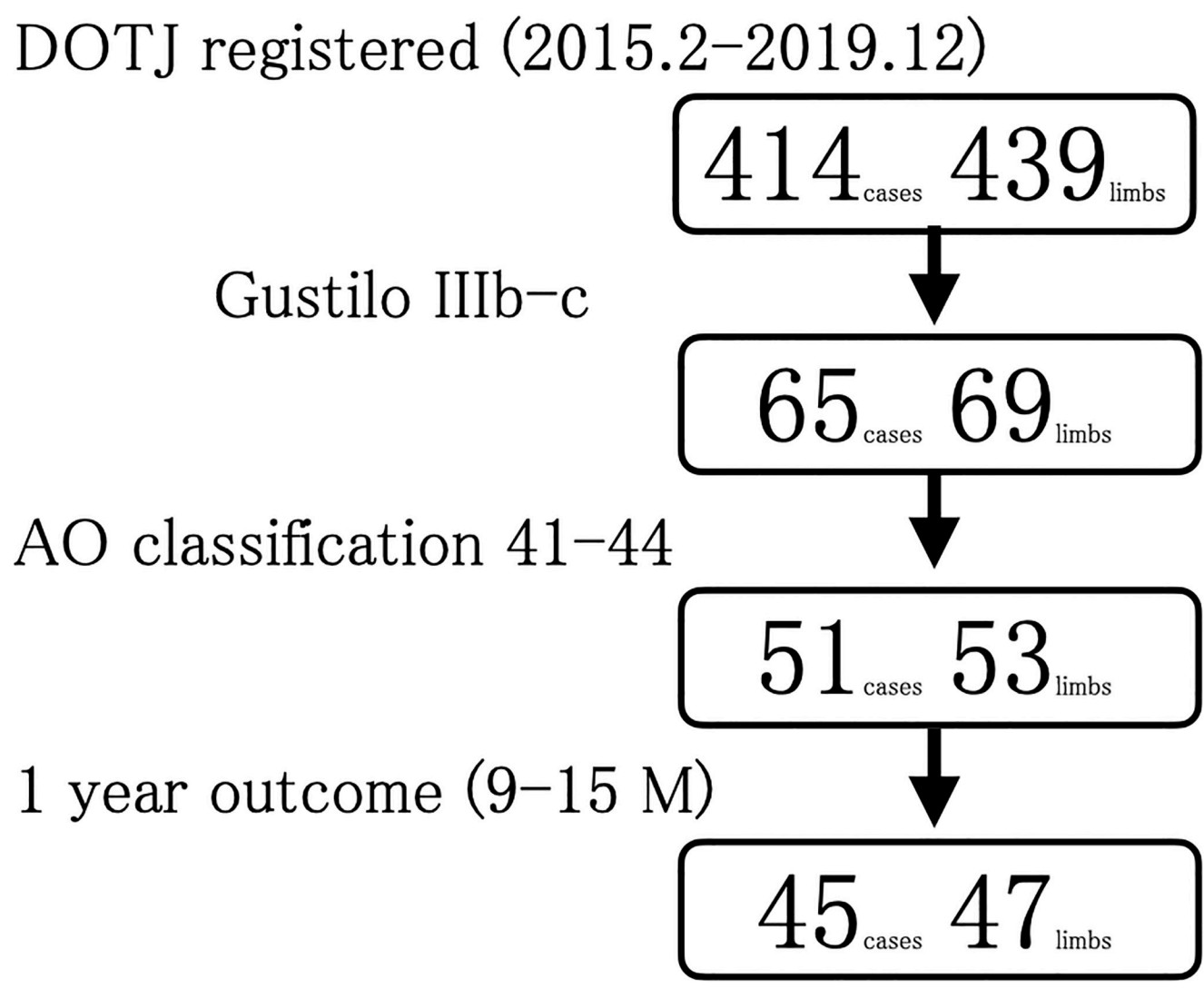

**Fig 1. Flow diagram of the study design.** DOTJ, Database of Orthopaedic Trauma managed by the Japanese Society for Fracture Repair.

(Fig 1). The follow-up rate was 88.2%. Patients were requested to provide self-reported outcomes 9–15 months after injury. Salvaging or amputating the injured limb was decided by the facility for each patient, depending on the extent of limb injury and the patient's general condition and social background. Of the 47 limbs of 45 patients, 34 limbs of 34 patients were salvaged, whereas 13 limbs of 11 patients were amputated. As an additional analysis with adjusted patient background, the same analysis procedures were carried out again after excluding patients with contralateral leg injuries, pelvic ring or acetabular fractures, and pre-injury functional impairment.

## Outcomes

The primary outcomes of our study were patient-reported outcomes based on LEFS and SF-8 questionnaires [8, 9]. The LEFS consists of 20 questions and is scored on a 0–80 points scale, with higher scores indicating higher functionality [8]. The SF-8 consists of eight health dimensions and two summary measures and is calculated using a national average of 50 points

(Norm-based Scoring), with higher scores indicating better health [9]. The secondary outcomes were the number of operations, surgical site infection rate, and patients' employment status (rate of change in employment), which are indicators of early recovery. The OTA-OFC summative score of the patients between the two groups was compared.

### Statistical analyses

Statistical analyses were conducted using the John's Macintosh Project software (Version 15.1.0; SAS Institute, Cary, NC, USA). Non-parametric testing with significance levels defined by a P-values of <0.05 was performed.

## Results

### Main result

Table 1 shows patient background characteristics, including the OTA-OFC summative score. Greater severity was observed in the amputation group than in the salvage group (14.3 and 9.8, respectively; P<0.001). Figs 2 and 3, and Table 1 show the study's primary outcomes. As shown in Figs 2 and 3, the mean LEFS score was 49.5 (range, 15–80) for the salvage group and 33.1 (range, 5–61) for the amputation group, suggesting that the salvage group showed significantly superior primary outcomes compared with those shown by the amputation group

**Table 1. Background characteristics of the patients, primary and secondary outcomes of the salvage and amputation groups.**

|  | Salvage | Amputation | P-values |
|---|---|---|---|
| Patients (limbs[a]) | 34 (34[a]) | 11 (13[a]) |  |
| Age (years) | 49.9 (7–75) | 49.0 (23–95) |  |
| Sex (M:F) | 28:6 | 7:4 |  |
| OTA-OFC score[a] | 9.8 (6–14) | 14.3 (8–15) |  |
| Pre-operative LEFS | 79.4 (66–80) | 72.7 (22–80) |  |
| Post-operative LEFS | 49.5 (15–80) | 33.1 (5–61) | 0.025* |
| SF-8 |  |  |  |
| Physical functioning | 41.8 | 34.6 | 0.076 |
| Role physical | 42.2 | 33.3 | 0.026* |
| Bodily pain | 49.2 | 44.5 | 0.107 |
| General health | 51.3 | 49.1 | 0.335 |
| Vitality | 50.9 | 46.8 | 0.076 |
| Social functioning | 44.8 | 38.5 | 0.060 |
| Role emotional | 45.1 | 39.4 | 0.050 |
| Mental health | 48.7 | 38.7 | 0.003* |
| Physical component summary | 43.8 | 39.0 | 0.073 |
| Mental component summary | 48.2 | 41.3 | 0.042* |
| Number of operations | 4.4 (2–8) | 2.3 (1–6) | 0.001* |
| Infection rate (%) | 35.3 | 38.5 | 1.000 |
| Rate of change in employment (%) | 26.5 | 18.2 | 0.705 |

[a] Number of injured limbs

Data presented as mean (range). OTA-OFC summative score was calculated for the number of injured limbs.

* indicates a significant difference.

LEFS, lower extremity functional scale; OTA-OFC, Orthopedic Trauma Association Open Fracture Classification; SF-8, Short-Form 8

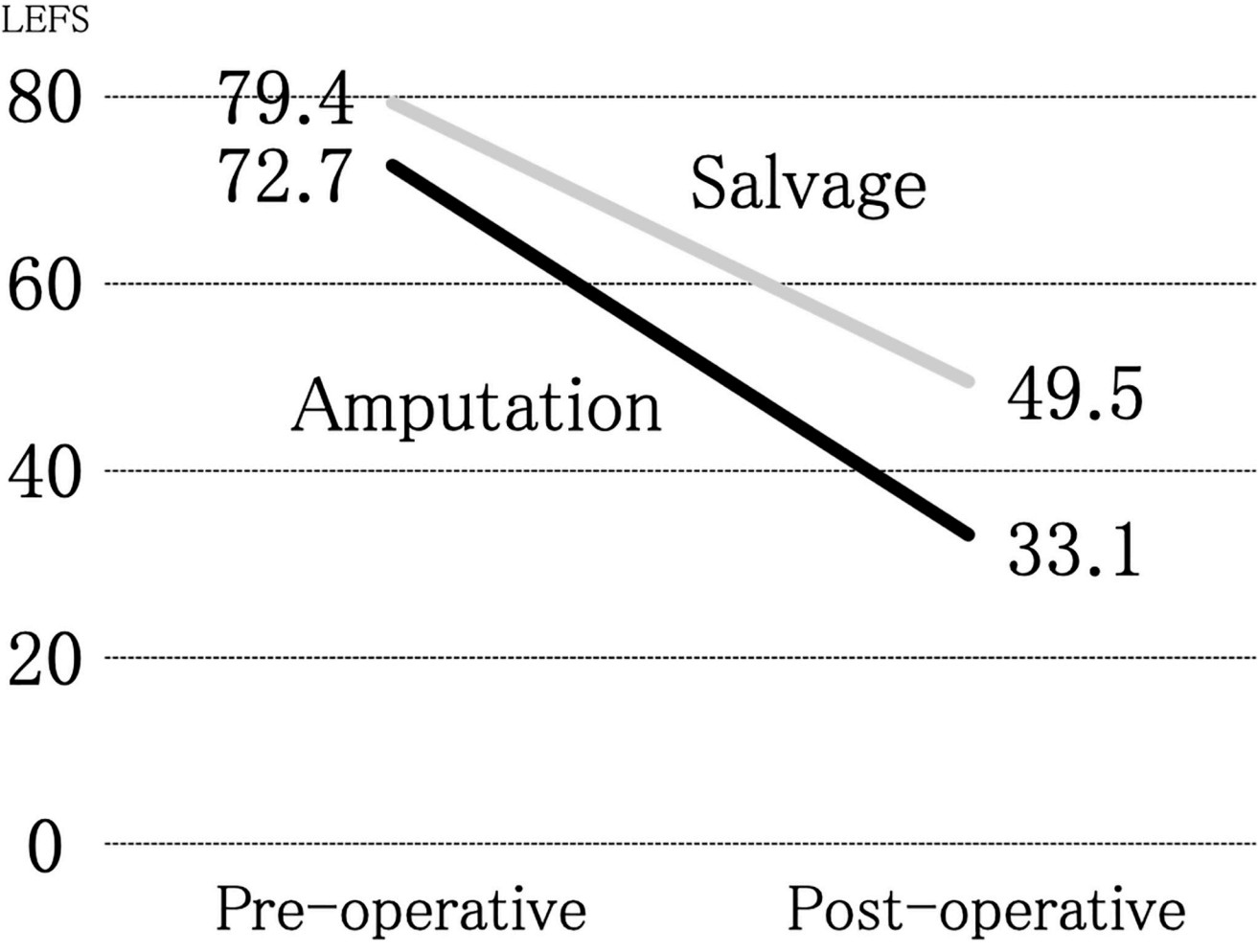

**Fig 2. Pre- and post-operative changes in lower extremity function scale (LEFS) scores of enrolled patients.**

(P = 0.025) (Table 1). In the SF-8, significant differences were observed between the salvage and amputation groups in the mean scores of the role–physical component (42.2 and 33.3; P = 0.026), mental health (48.7 and 38.7; P = 0.003), and mental component summary (48.2 and 41.3; P = 0.042). The limb salvage group showed better outcomes than the amputation group in all the aforementioned categories of the SF-8. Still, no significant differences were found between the two groups in the other categories of the SF-8 (Table 1).

Table 1 also shows the secondary outcomes of the study. Significant differences were noted in the number of operations between the salvage and amputation groups, with a mean of 4.4 (range, 2–8) and 2.3 (range, 1–6) (P = 0.001) operations, respectively. However, no significant differences in other secondary outcomes were noted between the two groups.

## Other analysis

Additional analysis was conducted for the 29 limbs of 29 patients after excluding 12 limbs of ten patients with contralateral lower limb injury, four limbs of four patients with pelvic ring or acetabular fractures, and two limbs of two patients with pre-injury LEFS scores of <77 (Fig 4). Hence, 24 patients in the salvage group and five in the amputation group were included in the additional analysis. The OTA-OFC summative scores were 10.1 and 14.6 for the salvage and

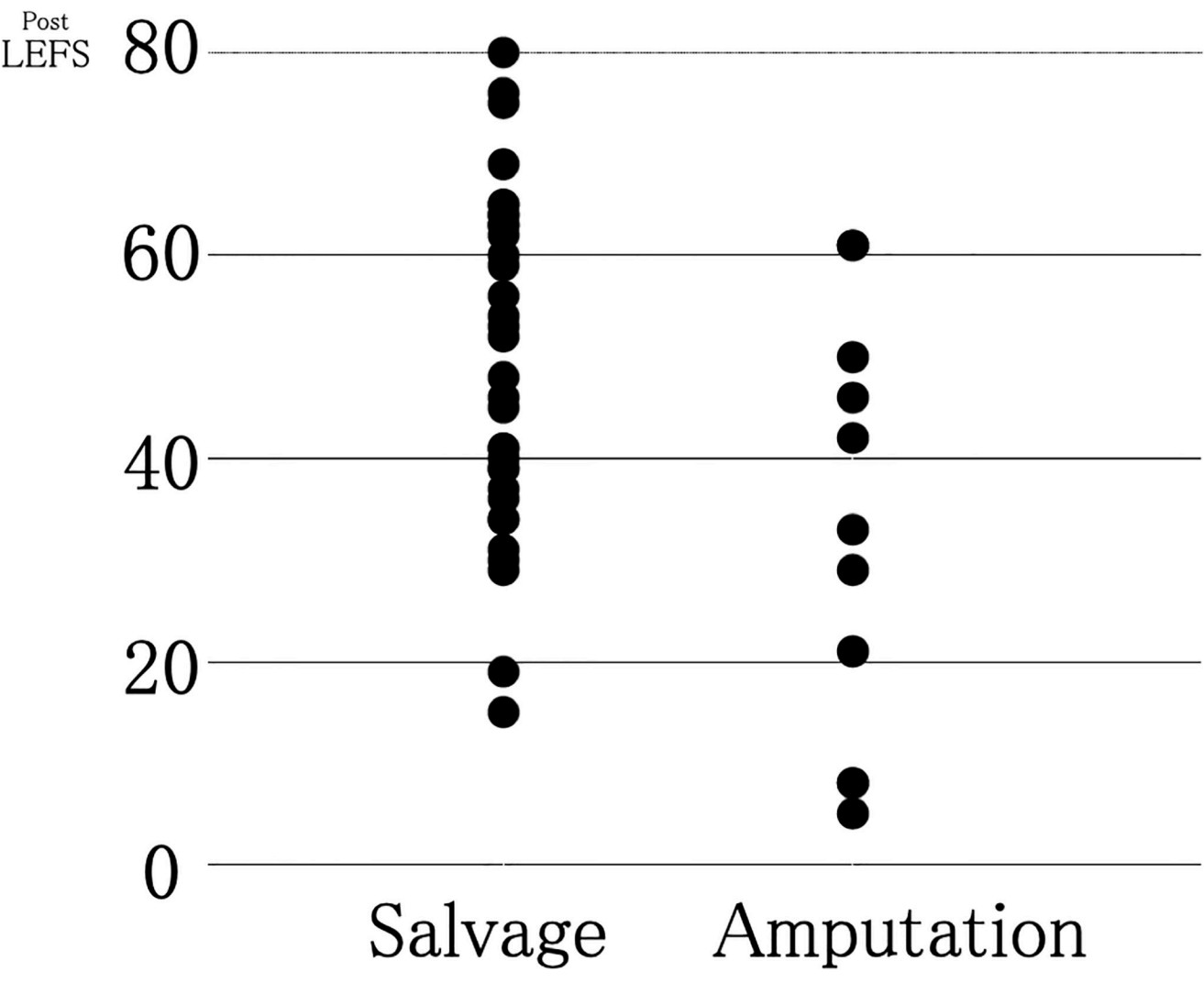

**Fig 3. Distribution of the lower extremity functional scale (LEFS) score of the enrolled patients.**

amputation groups, respectively. Severity was greater in the amputation group than in the salvage group (P = 0.001) (Table 2). No significant differences in LEFS scores between the salvage and amputation groups were noted. However, for the SF-8, the scores of bodily pain component (salvage, 49.0; amputation, 38.7; P = 0.022), vitality component (salvage, 51.7; amputation, 45.1; P = 0.036), mental health component (salvage, 48.4; amputation, 36.3; P = 0.005), and mental component summary (salvage, 48.2; amputation, 38.2; P = 0.012) were significantly different between the two groups. The salvage group showed better scores in all the aforementioned SF-8 categories than the amputation group (Table 2). No significant differences in secondary outcomes were noted between the two groups, except for the number of operations (salvage, 4.3 [range, 2–8]; amputation, 2.6 [range, 2–3]; P = 0.023) (Table 2).

## Discussion

### Key result

We initially believed that amputation contributed to the early recovery of the social life of patients with severe lower limb injuries. However, the obtained data on patient-reported

Patients

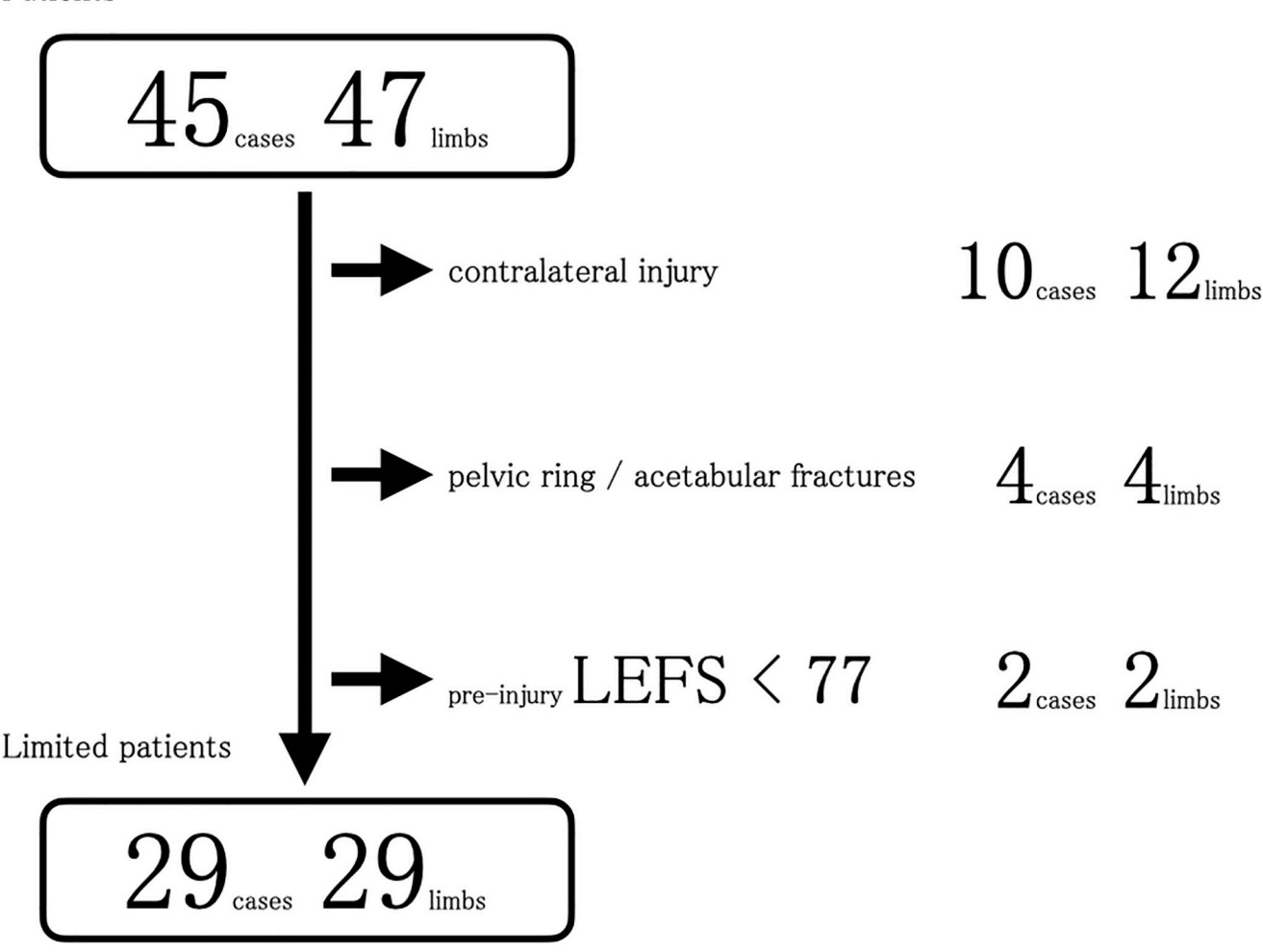

**Fig 4. Flow diagram of the additional analysis.** LEFS, lower extremity functional scale.

outcomes one year after injury, as investigated in this study, suggested that limb salvage provided better limb function and mental health outcomes. Additionally, while limb salvage required more operations, it did not affect the prevalence of surgical site infections and patient employment status. Furthermore, as shown in the LEFS score distribution in Fig 3, we found that the LEFS score in the amputation group did not exceed 62. The mean LEFS for the amputation group was 33.1, which was also inferior to healthy women in their 80s [8] (Fig 2). These findings implied that some functional restrictions remained in the amputation group, even in cases showing good progress.

Our study noted cases of co-manifestation of injuries in other body parts due to the severity of the primary injury. In addition, some patients had existing pre-injury functional impairment caused by aging, which affected the evaluation. Therefore, we conducted an additional analysis using the same method but for a limited number of patients, excluding patients with contralateral leg injuries, pelvic ring or acetabular fractures, and a pre-injury LEFS score of <77 (Fig 4). We established an LEFS score of <77 as the standard for pre-injury functional impairment because the median LEFS score in a study of healthy subjects was 77 [8]. Even

**Table 2. Characteristics of the patients included in the additional analysis, and results of the additional analysis for the primary and secondary outcomes.**

| | Salvage | Amputation | P-values |
|---|---|---|---|
| Patients | 24 | 5 | |
| Age (years) | 48.6 (7–73) | 37.4 (23–47) | |
| Sex (M:F) | 21:3 | 4:1 | |
| OTA-OFC score | 10.1 (7–14) | 14.6 (14–15) | |
| Pre-operative LEFS | 79.8 (78–80) | 80.0 (80) | |
| Post-operative LEFS | 50.4 (19–80) | 46.4 (33–61) | 0.686 |
| SF-8 | | | |
| Physical functioning | 41.7 | 38.4 | 0.645 |
| Role–physical | 42.2 | 35.7 | 0.296 |
| Bodily pain | 49 | 38.7 | 0.022* |
| General health | 51.9 | 46.3 | 0.085 |
| Vitality | 51.7 | 45.1 | 0.036* |
| Social functioning | 45 | 37.7 | 0.131 |
| Role emotional | 44.9 | 36.9 | 0.073 |
| Mental health | 48.4 | 36.3 | 0.005* |
| Physical component summary | 44 | 39.9 | 0.194 |
| Mental component summary | 48.2 | 38.2 | 0.012* |
| Number of operations | 4.3 (2–8) | 2.6 (2–3) | 0.023* |
| Infection rate (%) | 41.6 | 40.0 | 0.945 |
| Rate of change in employment (%) | 16.7 | 0.0 | 0.200 |

Additional analysis was performed after excluding patients with contralateral leg injuries, pelvic ring or acetabular fractures, and pre-injury LEFS scores of <77.

* indicates a significant difference. Data are presented as mean (range).

LEFS, lower extremity functional scale; OTA-OFC, Orthopedic Trauma Association Open Fracture Classification; SF-8, Short-Form 8

after limiting the analysis to fewer patients and considering patient backgrounds, no differences were observed in the functional aspects of patient-reported outcomes one year after injury, although the psychological aspects of the outcomes were better in the salvage group. However, considering the distribution of LEFS scores, while the amputation group did not perform well, some patients in the salvage group had poorer outcomes than all patients in the amputation group. In other words, not all patients in the salvage group had better outcomes than those in the amputation group (Fig 5). In the case of unsuccessful limb salvage surgery, the salvage group had worse functional and mental outcomes than the amputation group [5]. Variations in the outcomes of the salvage group cannot be overlooked.

Several reports [4–6, 10, 11] have suggested that amputation is better than limb salvage for treating severe lower limb injuries. This claim is based on various perspectives of rebuttals against treatments that aim for limb salvage, such as advances in prosthetic limb technology, perioperative complications, duration of rehabilitation, number of operations, rate of operation site infections, and financial problems. Limb amputation is often the intervention of choice based on these reports. A study reported that limb salvage is accompanied by long rehabilitation, higher total cost, and an increased likelihood of a larger number of additional operations and readmissions to the hospital [4]. Another study claimed that infection, reoperation, and hospitalization rates were significantly lower in the amputation group and that amputation provided better outcomes in terms of functioning and quality of life in patients with severe

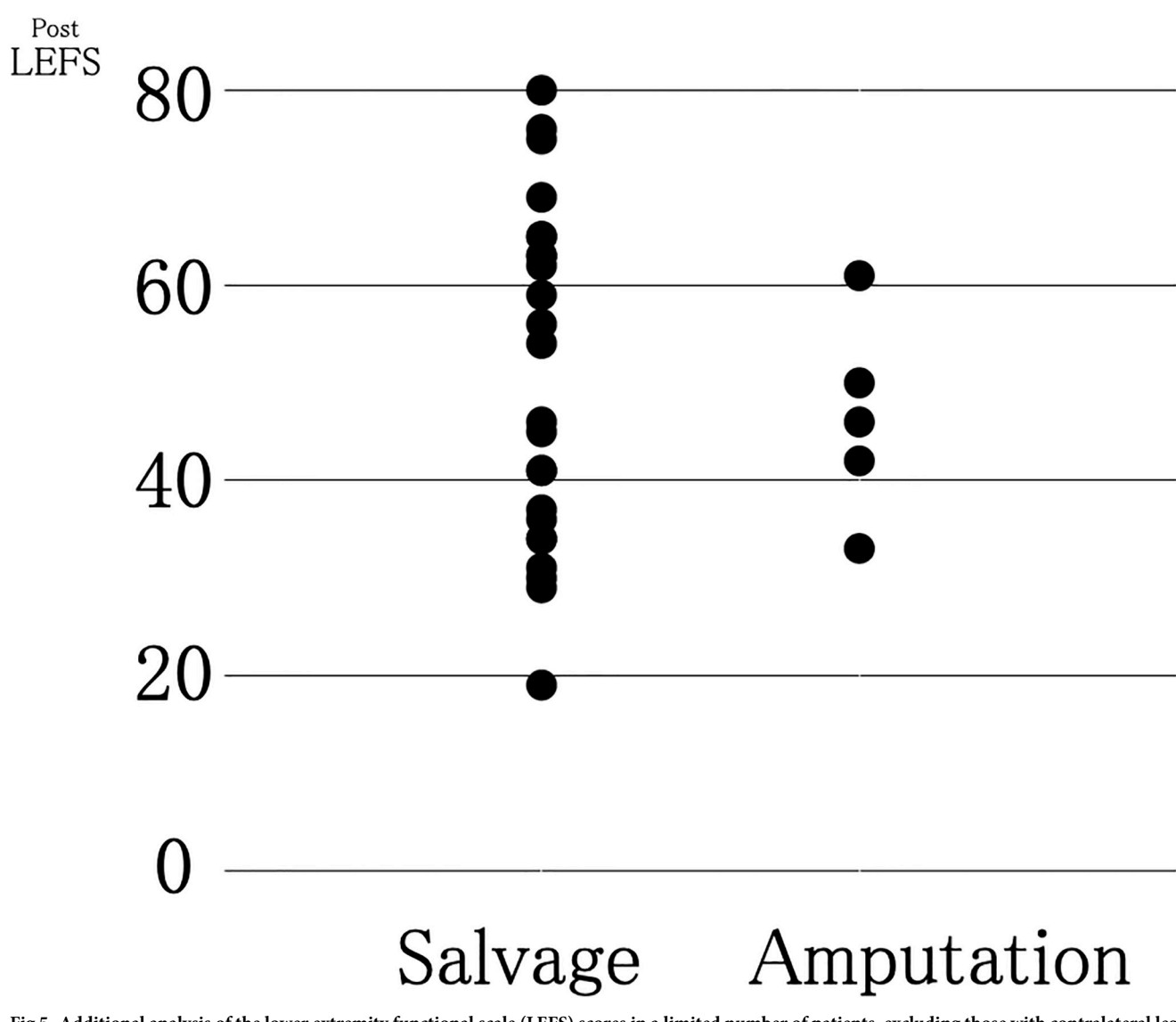

**Fig 5. Additional analysis of the lower extremity functional scale (LEFS) scores in a limited number of patients, excluding those with contralateral leg injuries, pelvic ring or acetabular fractures, and a pre-injury LEFS score of <77.**

leg injuries [6]. New prosthetic limb technology allowed amputees to participate in activities, exercise, and perform recreational activities that were previously impossible [10].

Some circumstances, such as those related to the patients' general condition and social background, necessitate amputation. However, according to several reports, limb salvage is more desirable than amputation in terms of psychological and long-term functional prognoses. Poor psychosocial outcomes after serious injuries have been reported by other investigators and may be associated with post-traumatic stress disorder [1]. In contrast, other reports have suggested that the time required by patients to return to work and the hospitalization period are similar between the two interventions [4–6, 12–14]. A meta-analysis showed that limb salvage and amputation are functionally equivalent, but limb salvage is the more psychologically acceptable approach [12]; this is consistent with another study claiming that amputation is more challenging to accept psychologically than limb salvage [1–3, 6]. Most patients

with salvaged limbs face problems daily due to the limited range of motion, but none want amputation as an intervention [14]. A previous prospective study showed no significant difference in the functional outcomes between the limb salvage and amputation groups for at least seven years [2, 3]. In the realm of evidence-based medicine, the LEAP studies provided a wealth of data but still failed to completely determine the treatment at the onset of severe lower extremity trauma [1–3].

It is becoming possible not only to preserve the injured limb but also to reconstruct it for better functional outcomes [5, 15–19]. One study showed an increased possibility of limb salvage, even in patients with popliteal artery injuries [15]. In contrast, another study reported that limb salvage is valuable even in patients with posterior tibial nerve injury [16]. Given these circumstances, the present study focused on early recovery based on the patients' perspectives.

## Limitation

One limitation of this study is the lack of randomization. According to the OTA-OFC summative score, patients who underwent limb amputation had more severe injuries. As amputation is ultimately chosen for mangled injuries, it is unavoidable for prospective studies. Therefore, we believe that randomized studies involving a larger number of patients are required, either by excluding patients with high OTA-OFC summative scores or by taking OTA-OFC summative scores into account. Furthermore, to improve the post-operative outcome of severe leg injuries, it is essential to properly implement a comprehensive rehabilitation program that includes the physical and psychosocial aspects [1–3, 10]. At our facility, we believe that each patient was provided the opportunity to undergo rehabilitation with appropriate duration and quality, regardless of whether the intervention chosen was limb salvage or amputation.

## Generalisability

With the advancement in reconstruction technology, it has become possible to salvage the affected limb. Therefore, it is crucial to understand the concept of "how the patient feels." We believe that the results of this study, which is based on patient-reported outcomes, are meaningful. It is desirable to acquire knowledge and skills in revascularization, microvascular surgery, and flap surgery for severe lower leg injuries and consider salvaging the affected limb. However, the results of our additional analysis on a limited number of patients revealed that some patients with limb salvage had lower functional ability than patients with amputation and that limb salvage conducted without ascertaining the condition completely may result in worse outcomes than those observed with amputation.

In conclusion, one year after lower limb injury, limb salvage resulted in better functional and mental health outcomes than amputation. As reconstruction technology has advanced and limb salvaging has become possible, the focus of studies should now be based on "how the patient feels;" hence, we believe that this study's results are based on patient-reported outcomes and are meaningful.

## Author Contributions

**Conceptualization:** Taketo Kurozumi.

**Data curation:** Taketo Kurozumi, Takahiro Inui, Yoshinobu Watanabe.

**Formal analysis:** Taketo Kurozumi.

**Investigation:** Taketo Kurozumi, Yuhei Nakayama, Akifumi Honda, Kentaro Matsui, Keisuke Ishii, Takashi Suzuki.

**Methodology:** Taketo Kurozumi, Takahiro Inui, Yoshinobu Watanabe.

**Project administration:** Taketo Kurozumi.

**Resources:** Taketo Kurozumi.

**Software:** Taketo Kurozumi.

**Supervision:** Taketo Kurozumi.

**Validation:** Taketo Kurozumi.

**Visualization:** Taketo Kurozumi.

**Writing – original draft:** Taketo Kurozumi.

**Writing – review & editing:** Taketo Kurozumi.

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
