## [Decision Letter · Decision Letter 0]

9 May 2022

PONE-D-22-06485Comparison of patient-reported outcomes at one year after injury between limb salvage and amputation: A prospective cohort studyPLOS ONE

Dear Dr. KUROZUMI,

Thank you for submitting your manuscript to PLOS ONE. After careful consideration, we feel that it has merit but does not fully meet PLOS ONE’s publication criteria as it currently stands. Therefore, we invite you to submit a revised version of the manuscript that addresses the points raised during the review process.

We look forward to receiving your revised manuscript.

Kind regards,

Arezoo Eshraghi, Ph.D.

Academic Editor

PLOS ONE

Journal Requirements:

“No. The funders had no role in study design, data collection and analysis, decision to publish, or preparation of the manuscript.”

Reviewers' comments:

Reviewer's Responses to Questions

**Comments to the Author**

1. Is the manuscript technically sound, and do the data support the conclusions?

Reviewer #1: Yes

2. Has the statistical analysis been performed appropriately and rigorously? 

Reviewer #1: I Don't Know

3. Have the authors made all data underlying the findings in their manuscript fully available?

Reviewer #1: Yes

4. Is the manuscript presented in an intelligible fashion and written in standard English?

Reviewer #1: Yes

5. Review Comments to the Author

Reviewer #1: Summary:

The authors present a small cohort study of patients with severe lower limb fractures, an uncommon but profoundly disabling injury. Deciding on the optimal treatment (salvage or amputation) for patients with these injuries is recognised as one of the most difficult decisions a surgeon can face. The results of this study advance understanding of patient centered outcomes following these injuries, and will contribute to the literature aiding decision-making between amputation and limb salvage.

Major issues:

Introduction & Discussion:

1) The Lower Extremity Assessment Project (LEAP) was a multi-centre, prospective cohort study between 1994 and 1997 that investigated the topic of this paper, and followed patients up at 2 (Bosse, NEJM) and 7 (MacKenzie, JBJS) years. The results of the LEAP study are now probably dated, and differ from the results of this study - however, the authors should report the key background knowledge on this topic in the introduction, and discuss their findings in the content of this previous work. Indeed, the authors suggest in the introduction that "prospective studies based on patient reported outcomes are extremely rare", yet the LEAP study is just this, and has provided the largest body of knowledge on the topic.

Methods:

1) Consider using the Strobe reporting guideline for cohort studies

Minor issues:

Introduction:

1) The introduction is short, well written and highlights a clear rationale for the study and potential knowledge gap. However, there are few references. Please could the authors consider referencing more of the statements if possible.

Results:

1) The magnitude of the functional disability in both groups is substantial, this in itself is an important finding, that is not highlighted. Could the authors consider showing this in graphical format (before - after LEFS scores)? And place in context what a LEFS score of 33 (average for amputees) means - on average moderate to quite a bit of difficulty with all activities of daily living!

Discussion:

1) Consider discussing the possible reasons for the considerable impact these injuries have on mental health ( a key finding of this paper)

6. PLOS authors have the option to publish the peer review history of their article (what does this mean?). If published, this will include your full peer review and any attached files.

Reviewer #1: No

---

## [Author Response · Author response to Decision Letter 0]

10 Jun 2022

Response to Reviewers

Journal Requirements:

Response: Thank you for the information. We have made the required changes.

Response: Thank you for the information. We have added the following to the title page. 

Request can be made to the Japanese Society for Fracture Repair secretariat (1-8-6, Shintomi Chuoku Tokyo 104-0041 Japan) or to jsfr@jsfr.jp and those who meet the criteria for access to anonymized patient level data will be granted data access.

“No. The funders had no role in study design, data collection and analysis, decision to publish, or preparation of the manuscript.”

Response: We have added the following to the cover letter and it has been added to the Title page.

The authors received no specific funding for this work.

Comments to the Author

5. Review Comments to the Author

Reviewer #1: Summary:

The authors present a small cohort study of patients with severe lower limb fractures, an uncommon but profoundly disabling injury. Deciding on the optimal treatment (salvage or amputation) for patients with these injuries is recognised as one of the most difficult decisions a surgeon can face. The results of this study advance understanding of patient centered outcomes following these injuries, and will contribute to the literature aiding decision-making between amputation and limb salvage.

Response: Thank you for understanding our arguments.

Major issues:

Introduction & Discussion:

1) The Lower Extremity Assessment Project (LEAP) was a multi-centre, prospective cohort study between 1994 and 1997 that investigated the topic of this paper, and followed patients up at 2 (Bosse, NEJM) and 7 (MacKenzie, JBJS) years. The results of the LEAP study are now probably dated, and differ from the results of this study - however, the authors should report the key background knowledge on this topic in the introduction, and discuss their findings in the content of this previous work. Indeed, the authors suggest in the introduction that "prospective studies based on patient reported outcomes are extremely rare", yet the LEAP study is just this, and has provided the largest body of knowledge on the topic.

Response: Thank you for this suggestion. We have added its content in the background and discussion. Page 3, Line 65-71; Page 14, Lines 288-290

Methods:

1) Consider using the Strobe reporting guideline for cohort studies

Response: Thank you for your suggestion. We have rewritten some of them to conform to the guidelines. Page 3, Lines 79, 81; Page 4, Lines 86, 89.

Minor issues:

Introduction:

1) The introduction is short, well written and highlights a clear rationale for the study and potential knowledge gap. However, there are few references. Please could the authors consider referencing more of the statements if possible.

Response: Thank you very much. We have added some additional references and background information there. Page 3, Lines 65, 66, 68, 71.

Results:

1) The magnitude of the functional disability in both groups is substantial, this in itself is an important finding, that is not highlighted. Could the authors consider showing this in graphical format (before - after LEFS scores)? And place in context what a LEFS score of 33 (average for amputees) means - on average moderate to quite a bit of difficulty with all activities of daily living!

Response: Thank you for this suggestion. We have added its content in the result and discussion. Pages 10-11, Lines 176-206; 217-218

Discussion:

1) Consider discussing the possible reasons for the considerable impact these injuries have on mental health ( a key finding of this paper)

Response: Thank you for this insightful comment. We have added a discussion around this topic to the discussion section. Page 13, Lines 275-278

---

## [Decision Letter · Decision Letter 1]

1 Jul 2022

PONE-D-22-06485R1Comparison of patient-reported outcomes at one year after injury between limb salvage and amputation: A prospective cohort studyPLOS ONE

Dear Dr. KUROZUMI,

Thank you for submitting your manuscript to PLOS ONE. After careful consideration, we feel that it has merit but does not fully meet PLOS ONE’s publication criteria as it currently stands. Therefore, we invite you to submit a revised version of the manuscript that addresses the points raised during the review process.

We look forward to receiving your revised manuscript.

Kind regards,

Arezoo Eshraghi, Ph.D.

Academic Editor

PLOS ONE

Journal Requirements:

Reviewers' comments:

Reviewer's Responses to Questions

**Comments to the Author**

1. If the authors have adequately addressed your comments raised in a previous round of review and you feel that this manuscript is now acceptable for publication, you may indicate that here to bypass the “Comments to the Author” section, enter your conflict of interest statement in the “Confidential to Editor” section, and submit your "Accept" recommendation.

Reviewer #1: (No Response)

2. Is the manuscript technically sound, and do the data support the conclusions?

Reviewer #1: Yes

3. Has the statistical analysis been performed appropriately and rigorously? 

Reviewer #1: I Don't Know

4. Have the authors made all data underlying the findings in their manuscript fully available?

Reviewer #1: Yes

5. Is the manuscript presented in an intelligible fashion and written in standard English?

Reviewer #1: Yes

6. Review Comments to the Author

Reviewer #1: Minor issue:

Thank you for including reference to the LEAP study body of work.

This was a large multi-centre, prospective observational study, that took place between 1994 and 1997, and remains one of the most important civilian studies on the topic.

In the revised introduction, the sentence "Moreover, many previous investigations on this topic are retrospective cohort studies or meta-analyses on retrospective studies, such as the LEAP study, whereas prospective cohort studies based on patient-reported outcomes are extremely rare [1-3]." implies that the LEAP study is a retrospective cohort study or a meta-analysis of retrospective studies, which it is neither. Please consider rewording to something like: "Moreover, many previous investigations on this topic are retrospective cohort studies or meta-analyses on retrospective studies, whereas prospective cohort studies based on patient-reported outcomes, such as the LEAP study, are extremely rare [1-3]."

7. PLOS authors have the option to publish the peer review history of their article (what does this mean?). If published, this will include your full peer review and any attached files.

Reviewer #1: No

---

## [Author Response · Author response to Decision Letter 1]

3 Jul 2022

6. Review Comments to the Author

Reviewer #1: Minor issue:

Thank you for including reference to the LEAP study body of work.

This was a large multi-centre, prospective observational study, that took place between 1994 and 1997, and remains one of the most important civilian studies on the topic.

In the revised introduction, the sentence "Moreover, many previous investigations on this topic are retrospective cohort studies or meta-analyses on retrospective studies, such as the LEAP study, whereas prospective cohort studies based on patient-reported outcomes are extremely rare [1-3]." implies that the LEAP study is a retrospective cohort study or a meta-analysis of retrospective studies, which it is neither. 

Please consider rewording to something like: "Moreover, many previous investigations on this topic are retrospective cohort studies or meta-analyses on retrospective studies, whereas prospective cohort studies based on patient-reported outcomes, such as the LEAP study, are extremely rare [1-3]."

Response: Thank you for this suggestion. We have made the required changes. Page 5, Line 68-69

---

## [Editor Report · Decision Letter 2]

6 Sep 2022

Comparison of patient-reported outcomes at one year after injury between limb salvage and amputation: A prospective cohort study

PONE-D-22-06485R2

Dear Dr. Kurozumi,

We’re pleased to inform you that your manuscript has been judged scientifically suitable for publication and will be formally accepted for publication once it meets all outstanding technical requirements.

Kind regards,

Alvan Ukachukwu, *MD, MSc.GH*

Academic Editor

PLOS ONE
---

## [Editor Report · Acceptance letter]

9 Sep 2022

PONE-D-22-06485R2 

Comparison of patient-reported outcomes at one year after injury between limb salvage and amputation: A prospective cohort study 

Dear Dr. Kurozumi:

I'm pleased to inform you that your manuscript has been deemed suitable for publication in PLOS ONE. Congratulations! Your manuscript is now with our production department. 

Kind regards, 

on behalf of

Dr. Alvan Ukachukwu 

Academic Editor

PLOS ONE